# Nigerian Muslim’s Perceptions of Changes in Diet, Weight, and Health Status during Ramadan: A Nationwide Cross-Sectional Study

**DOI:** 10.3390/ijerph192114340

**Published:** 2022-11-02

**Authors:** Sahabi Kabir Sulaiman, Fatimah Isma’il Tsiga-Ahmed, MoezAlIslam E. Faris, Muhammad Sale Musa, Udoyen Abasi-okot Akpan, Abdullahi Muhammad Umar, Salisu Maiwada Abubakar, Kholoud K. Allaham, Taif Alyammahi, Munther A. Abdbuljalil, Syed Fahad Javaid, Moien AB Khan

**Affiliations:** 1Department of Internal Medicine, Yobe State University Teaching Hospital, Damaturu P.M.B 1047, Nigeria; 2Department of Community Medicine, Bayero University Kano/Aminu Kano Teaching Hospital, Kano P.M.B 3452, Nigeria; 3Department of Clinical Nutrition and Dietetics, College of Health Sciences, University of Sharjah, Sharjah P.O. Box 26666, United Arab Emirates; 4Department of Medicine, National Pirogov Memorial Medical University, 21018 Vinnytsia, Ukraine; 5Department of Medicine, Federal College of Education, Kano P.M.B 01000, Nigeria; 6Department of Biochemistry and Africa Centre of Excellence for Population Health and Policy, Bayero University, Kano P.M.B. 3011, Nigeria; 7Health and Wellness Research Group, Department of Family Medicine, College of Medicine and Health Sciences, United Arab Emirates University, Al-Ain P.O. Box 15551, United Arab Emirates; 8Health and Wellness Research Group, Department of Psychiatry and Behavioral Sciences, College of Medicine and Health Sciences, United Arab Emirates University, Al-Ain P.O. Box 15551, United Arab Emirates; 9Primary Care, NHS North West London, London TW3 3EB, UK

**Keywords:** eating habits, intermittent fasting, Nigeria, Ramadan fasting, sustainable development goals, COVID-19

## Abstract

The Islamic month of Ramadan is marked by fasting, when Muslims refrain from eating and drinking from dawn to sunset, which has an impact on their dietary habits. The study aimed to assess Nigerian Muslims’ dietary modifications during Ramadan and their related changes in body weight and health status. A web-based cross-sectional study was conducted among Nigerian adult Muslims. The survey assessed sociodemographic, dietary habits, eating behaviors, food choices, perceived weight changes, and health status. The logistic regression model was used to assess the predictors of weight change and perceived health status. There were 770 participants, 62.9% of whom were women, ranging in age from 18 to 60 years with a mean age of 27.7 ± 6.4 years. Fruits, palm dates, homemade foods, milk products, and vegetables were more frequently consumed. There were fewer energy drinks, pastries, salty snacks, and carbonated or sugared drinks consumed during Ramadan than before. Over half (54.6%, 95% CI: 51.0–58.9%) of the respondents lost weight during Ramadan, 37.0% (95% CI: 17.4–38.6%) maintained their weight and 8.4% (95% CI: 6.6–10.6%) gained weight during the month. Nearly all (97.3%, 95% CI: 95.8–98.3%) reported having good health during Ramadan, and 2.7% (95% CI: 1.7–4.1%) reported having a poorer health state during Ramadan. There was a significant weight loss and healthy dietary change associated with Ramadan fasting in Nigeria. Public health measures must be in place to impart such positive health behaviors so that such healthy habits continue throughout the year.

## 1. Introduction

Muslim fasts from dawn to dusk for 29 or 30 days during Ramadan, the ninth month of the Islamic lunar calendar [1]. Ramadan intermittent fasting (RIF) varies between 12 and 20 hours per day based on latitude [2]. RIF involves Muslims eating a predawn meal called Suhoor then abstaining from eating, drinking, and engaging in sexual activity until sunset, when they break their fast with eating and drinking (Iftar) [3]. There is a profound direct and indirect effect of RIF on health, and these effects manifest in physical, metabolic, immunologic, and mental aspects of health [4]. Though there are few discrepancies with the current evidence on the impact of intermittent fasting (IF) and RIF on health [4,5], there is an overwhelming report of general health improvement [5,6]. 

Evidence suggests that RIF has a therapeutic role on biological indicators of various noncommunicable diseases (NCDs) [7,8,9], including heart diseases, stroke, cancer, diabetes, and chronic respiratory disease. Such NCDs have been shown to share common modifiable behavioral risk factors such as tobacco use, unhealthy diet, lack of physical activity, and the harmful use of alcohol that ultimately lead to overweight and obesity, raised blood pressure, and raised cholesterol [10]. With RIF, high-density lipoproteins (HDLs) [11,12,13] were shown to increase, while fasting blood glucose (FBG) [4,14], triglycerides (TG) [4,12,13], low-density lipoproteins (LDLs) [4,12], and total cholesterol (TC) [4,12] were shown to decrease. A reduction in blood pressure (BP) [11,15] with improvement in anthropometric parameters as evidenced by a decrease in the BMI [16] and waist circumference (WC) [11] has also been widely reported. Similarly, lowering the serum levels of the proinflammatory cytokines IL-6, IL-1β, and TNF-α RIF significantly contributes to immunomodulation [17,18,19]. In addition, RIF also plays a role in healthy longevity, thus slowing aging and increasing life expectancy [17,18]. Though RIF has been reported to cause weight loss, such weight loss along with changes in favorable biochemical parameters are only possible depending on the quality of the diet associated with physical activity [5,20] 

We aim to identify the impact of RIF on health- and dietary-related behaviors. While there exist many different studies on non-Ramadan dietary patterns in Nigeria [21,22,23] and few studies regarding the health benefits of Ramadan in Muslim Ramadan-fasting Nigerians [24,25], literature is scarce with regard to dietary patterns brought by the fasting and how that is reflected in alterations in the body weight or BMI and personal health state. We hypothesized that various nutritional changes might affect the body weight and health status due to fasting during Ramadan during the COVID-19 pandemic. The sustainable development goals (SDGs) (SDG3—good health and well-being) adopted by the United Nations (UN) in 2015 pledge to deliver healthy lives and well-being across all ages by 2030 [26]. The coronavirus pandemic in 2019 (COVID-19) [27] has made this achievement more challenging due to restrictions on mobility, financial crisis, lack of food security, and inability to obtain quality nutrition [27,28,29]. We further hypothesized that RIF has impacted health and related behaviors and SDG3. The aim of the study was to examine the diet modifications during Ramadan and their association with self-reported health status and body changes among Nigerian adults.

## 2. Methods

### 2.1. Study Design, Setting, and Participants

A nationwide cross-sectional web-based survey was conducted between 9 May 2021 and 4 June 2021 among Nigerian adults who fasted during Ramadan. These dates correspond to the end of Ramadan and the following four weeks after Ramadan. Healthy Nigerian Muslims residing in Nigeria at the time of the survey, aged 18 years and above, who were able to comprehend and write in English, were invited to participate in the study. Those who were under 18 years old or nonresidents in Nigeria were excluded. 

### 2.2. Ethics

This study was conducted in accordance with the Declaration of Helsinki [30] and was approved by the research ethics committee of the Yobe State University Teaching Hospital, Damaturu, Nigeria (YSUTH/MAC/EC/02), and the Social Sciences Research Ethics Committee (REC) of the United Arab Emirates University (Approval Number ERS_2021_7308). Informed consent was obtained from all respondents, and participation was voluntary. 

### 2.3. Study Instrument and Data Collection

Convenience and snowball sampling methods were employed. Data were anonymously collected with no identifying information using single pre-tested, structured, self-administered, web-based, electronic questionnaires from previous similar studies [31,32,33,34]. The survey was created from previous validated questionnaires [31,32,33,34] and distributed through social media platforms and email through a unique Google Form link (docs.google.com/forms). In order to determine the reliability of the questionnaire, Cronbach’s alpha was calculated. A Cronbach’s alpha of ≥0.80 indicated that the measures in the questionnaire were reliable.

### 2.4. Sample Size

Fisher’s formula was used to estimate the sample size for this study [35]. Taking into consideration the primary outcomes of weight change and health status, two estimates were made. Based on the proportion of respondents who lost weight during Ramadan in a previous study [36], 95% confidence level, and 5% margin of error, a minimum sample size of 363 was obtained. A 10% increase was made in order to adjust for nonresponses, and was rounded up to 400. In the second sample size calculation, we used the percentage of respondents who were satisfied with their health status (30.6%) from study in a similar setting [37], 95% confidence level, and a 5% error margin. We estimated a minimum sample size of 384, which was increased by 10% to account for nonresponses and rounded up to 423. 

### 2.5. Description of Measures

The survey questionnaire elicited information from the respondents about their sociodemographic characteristics, eating behavior with dietary diversity, weight change, and health status before and during Ramadan.

### 2.6. Sociodemographics

Survey participants were asked about their age; sex; residence (rural or urban); marital status; the highest level of education; occupational status (employed or unemployed); work sector (private or government); father’s and mother’s level of education; total household income; household income with reference to the minimum household expenditure of NGN 137,600 (equivalent to USD 354) [38,39] for families in Nigeria; living status before and during the month of Ramadan (alone or with family); personal and family history of obesity, diabetes, hypertension, or heart disease; and number of days fasted during Ramadan.

### 2.7. Dietary Habits, Eating Behaviors, and Food Choices

Information about the participants’ diet and modified eating practices since the beginning of fasting (quality, quantity, type of products, mealtime schedule) was elicited using questionnaires. The dietary diversity section inquired about the modification of common food groups: vegetables, fruits, cereals and grains, oils and fats, milk and milk products, pulses, dates, fish and seafood, low-fat meat (chicken and turkey), sugar, salty snacks, fried foods, carbonated or sugary drinks, energy drinks, hot beverages, pastries, homemade foods, traditional foods, and fast foods; and how many times fast food and eating out were ordered before and during the month of Ramadan based on the short food frequency questionnaire [31,33]. Likewise, eating behavior before and during the fasting month of Ramadan was assessed by asking questions about the quantity of water drunk per day, snacking, consuming large quantities of food, and eating despite not feeling hungry [40,41]. 

## 3. Assessment of Weight, Height, Perceived Weight Changes, and Health Status

Respondents were asked to self-report their current body weights (in kilograms) and heights (in centimeters), which were used to calculate their corresponding BMI (kg/m^2^) during the analysis and classified into four categories: normal (18.5–24.9), underweight (<18.5 kg/m^2^), overweight (25.0–29.9), and obese (>30 kg/m^2^). Respondents were also asked about their perceived weight status (lost, maintained, or gained) during Ramadan; self-perceived current weight status (normal, underweight, overweight, and obese); and the amount of weight perceived to have been lost (−1 to −4.5 or more), maintained (0), or gained (+1 to +4.5 or more) during Ramadan. Respondents were similarly asked to grade their overall health status before and during Ramadan on a Likert scale (poor, fair, good, very good, excellent), which were later amalgamated into either poor (fair and poor) or good (good, very good, and excellent) during the analysis.

### Data Analysis

All analyses were carried out using STATA version 15.0 (StataCorp LLC, College Station, TX, USA). Mean and standard deviation (SD) were used to summarize the continuous variables, such as the age of the respondents; whereas income, being a skewed variable, was presented in median and interquartile range (IQR). Frequencies and percentages were obtained for categorical variables such as eating behaviors and food groups consumed during Ramadan. Pearson’s chi-square or Fisher’s exact test was used to assess the differences in eating behaviors before and during Ramadan. Proportions of respondents with different levels of weight change and health state during Ramadan were calculated with their corresponding 95% confidence intervals (CIs). Only individuals who knew their pre-Ramadan weight and those who fasted were included in the analysis. This was followed by cross-tabulation of explanatory variables with classes of weight change and health state to explore unadjusted associations. Using a forward modeling approach, binary logistic and multinomial logistic regressions were used for multivariable analyses of risk factors for weight change and health state categories, respectively. Sex, BMI, and age were considered a priori confounding variables for all models as identified from the previous literature [42]. Starting with a priori confounders in the base model, independent variables with *p* < 0.10 at the bivariate level were sequentially included in the logistic regression models adding the potential confounders one at a time based on the strength of evidence obtained. A final model was built with those variables that changed the odds ratio by at least 10%. Crude and adjusted odds ratios (CORs and AORs, respectively) with their respective 95% confidence intervals were calculated, and type I error was fixed at 5% for all tests. 

## 4. Results

A total of 770 respondents participated in the study, and their ages ranged from 18 to 60 (27.7± 6.4) years. The majority (73.0%, *n* = 562) lived in the urban area, and 62.9% (*n* = 484) were women. About two-thirds (66.0%, *n* = 508) were university students, and 66.2% (*n* = 510) were married. Approximately half (51.8%, *n* = 399) were unemployed, and their family monthly income ranged from NGN 5000 to NGN 6,000,000, with a median income and interquartile range (IQR) of NGN 100,000 (NGN 50,000, NGN 300,000). Almost a quarter of the respondents (24.2%, *n* = 186) were obese; 12.7% (*n* = 98) had a history of diabetes, hypertension, or heart disease; and a sizeable number (60.7%, *n* = 467) had a family history of hypertension, diabetes, or heart disease. Most of the respondents (83.0%, *n* = 639) fasted for between 21 and 30 days, and more than one-third (38.2%, *n* = 294) snacked between once and twice from Iftar to Suhoor every day during Ramadan (Table 1). 

### 4.1. Diet Pattern and Diversity during Ramadan 

The food groups with the most increased consumption during Ramadan were fruits (72.4%, *n* = 540) and dates (70.5%, *n* = 511). Consumption of homemade food was increased by 38.5% (*n* = 283), milk and milk products by 38.4% (*n* = 278), and vegetables by 33.3% (*n* = 243) among the respondents. Almost half of the respondents reduced intake of energy drinks (44.6%, *n* = 201), pastries (44.0%, *n* = 267), salty snacks (42.0%, *n* = 266), and carbonated or sugary drinks (40.5%, *n* = 264). Intake of most food types was maintained by more than half of the respondents; 55.9% (*n* = 421) maintained their pre-Ramadan intake of red meat, 55.8% (*n* = 394) maintained fish intake, and 55.1% maintained intake of traditional foods and low-fat red meat. Over two-thirds of the respondents did not change their intake of pulses (69.6%, *n* = 434), salt (68.1, *n* = 498), and fats (59.4%, *n* = 443) during Ramadan (Figure 1). 

Table 2 shows that the improvement in self-evaluated quality of diet consumed by the respondents (*p* = 0.005) and eating despite not being hungry (*p* < 0.001) were significantly higher during Ramadan. However, respondents significantly consumed lower quantities of water during Ramadan (*p* = 0.03). While 23.8% (*n* = 183) of the respondents drank between one and three cups per day before Ramadan, 29.0% (*n* = 223) drank a similar quantity during Ramadan. Additionally, the frequencies of eating out, taking takeout, or snacking were higher before Ramadan. Respondents significantly reduced these habits during the fasting period (*p* < 0.001). 

### 4.2. Weight Change and Health State during Ramadan

During Ramadan (Figure 2), over half of the respondents (54.7%, 95% CI: 50.7–58.7) lost weight, 37.0% (95% CI: 33.2–40.9%) maintained their weight, and 8.3% (95% CI: 6.3–10.7%) gained weight. During Ramadan, the vast majority (97.3%, 95% CI: 95.8–98.3%) reported good health, whereas 2.7% (95% CI: 1.7–4.1%) reported poorer health.

### 4.3. Predictors of Weight Change and Health State during Ramadan

Table 3 shows that snacking and eating out were significant predictors of weight gain during Ramadan. Respondents who ordered a takeaway between once and twice weekly were almost four times more likely to gain weight than those who rarely or never ordered a takeout (AOR: 3.7, 95% CI: 1.4–9.8). Furthermore, the odds of gaining weight during Ramadan among respondents who ate fewer quantities of food were almost three times higher than those who consumed more significant amounts of food (AOR: 2.6, 95% CI: 1.2–5.9). We found no variables to predict weight loss during Ramadan. 

Regarding perceived health state during Ramadan, occupation, snacking, and diet quality were associated with the health state of respondents at the bivariate level. Nonetheless, there was insufficient evidence to show that they were independent predictors of health state during Ramadan (Table 4).

## 5. Discussion

The changes observed in the reported body weight and overall perceived health status of the respondents reflect the extensive dietary and lifestyle modifications adopted by Muslims fasting in Ramadan in Nigeria. To the best of our knowledge, this is the first nationwide study performed to assess Ramadan dietary patterns in Nigeria concerning the changes in body weight and self-reported overall health status. Among the food types that increased were fruits (72.4%) and dates (70.5%), having the highest consumption increments, followed by milk and milk products by 38.4%, and vegetables by 33.3%. Similarly, a marked increase in the consumption of fruits, milk, and milk products was reported among Ghanaian adolescents fasting during the holy month of Ramadan, although consumption of dark-green leafy vegetables was reported to have decreased [2]. Similarly, in Iran, fruit consumption was above the recommended range, whereas that of vegetables was much below the recommended range [43]. In a year-round Lebanese study, it was found that the intakes of cereals, cereal-based products, pasta, eggs, nuts and seeds, milk and dairy, as well as fats and oils were decreased.

On the other hand, intakes of vegetables, dried fruit, Arabic sweets, cakes and pastries, and sugar-sweetened beverages were considerably increased during Ramadan when compared with the remaining non-Ramadan periods of the year (*p* < 0.05) [44]. One of the possible explanations for healthy behaviors during Ramadan could be that Ramadan happened in the wake of the pandemic, and there was evidence and advice with regard to having healthy nutritious food during the COVID-19 outbreak from various public organizations and government bodies, which may have influenced healthy behavior [45,46]. Furthermore, with the current pandemic, taking portions of common and less expensive fruits and dates while breaking fast and limiting portions at mealtime, consuming less expensive or less preferred foods, could have altered the household food consumption [23]. Nevertheless, the food types that the surveyed population had maintained the highest intake of were pulses, salt, and fats. Other food types with maintained consumption by over 50% include cereals, homemade foods, and traditional foods, which in Nigeria are mostly the same as cereals and tubers. In addition, the consumption of most protein-rich foods was maintained by slightly more than half of the respondents, including red meat by 55.9%, fish and other kinds of seafood by 55.8%, and low-fat red meat by 55.1%. 

We found a marked reduction (nearly half) in the consumption of certain food types, including energy drinks by 44.6%, pastries by 44%, salty snacks by 42%, and carbonated and sugary beverages by 40.5%. Just as in this study, similar studies in many other populations reported neither an increase nor decrease in the usual energy consumption during Ramadan [40,47]. Similarly, reduced energy intake during Ramadan has been reported in several populations fasting during Ramadan globally [41,48,49,50]. In contrast, studies have also reported an otherwise increased energy consumption [51,52]. Notably, despite the varied picture we obtained in this study about dietary modification, the self-evaluated quality of diet consumed by the participants in the survey (*p* = 0.005) was significantly higher during Ramadan, with an overwhelming 92.7% reporting a good quality diet during the fasting period. Many of the respondents (30.4%) reported having a family income above the country’s minimum wage of NGN 137,60, enabling the purchase of high-quality food varieties. However, 37.5% of respondents were not aware of their family’s monthly income. Ramadan is also a period of charity, feeding, and almsgiving [53], which could also explain this highly reported improved quality of diet by the survey respondents, even though 32.4% of them have a gross family monthly income that is less than the country’s minimum. Importantly, we found that being employed, infrequent snacking, and improved self-evaluated quality of diet consumed by the respondents were associated with improved self-reported health status at the bivariate level, although not independent predictors of health state during Ramadan. Many factors could, independently or in combination, account for the vast differences in the results of these studies, including cultural differences, seasonality, and the pandemic.

Eating out, having takeaways, and snacking were significantly higher before Ramadan when compared with during Ramadan. Weight gain was nearly fourfold among respondents who ordered a takeaway once or twice per week than those who rarely or never did that. In contrast to favorable eating behaviors, the participants were eating despite not feeling hungry and reducing quantities of water intake during Ramadan. Despite this, many of the participants reported weight loss during Ramadan, although we found no predictors of weight loss based on the logistic regression of the variables at hand.

Studies carried out in different continents have widely reported this weight loss during Ramadan among populations fasting during Ramadan, both in normal weight [54,55,56] and overweight or obese individuals [57,58]. Significant sex disparity in weight loss during RIF has also been reported among normal-weight individuals, with men having almost twice the weight loss as women [16]. On a similar line, a study conducted on male students in Malaysia showed a reduction in body weight, and significant improvement in the waist-to-hip ratio was observed [59]. More than one-third (37.0%) of the respondents in this study reported a stable weight during Ramadan. Our results are similar with other studies that reported no significant change in weight during Ramadan in normal-weight individuals [57,58] and overweight or obese people [60]. More than 10% self-reported weight gain in our study. The study findings are similar with a study conducted in Saudi Arabia, where three in five of the respondents’ self-reported weight gain was attributed to decreased physical inactivity and increased food intake [20]. 

We found that more than one–third are overweight or obese, which is alarming because most of our respondents are from younger age groups falling between 21 and 30 years of age. Unless modified, they carry these risk factors as they transition to future decades, which predispose them to endemic NCDs such as cardiovascular diseases, diabetes, and cancer. However, even more alarming is recalling that over 1 in 10 (12.7%) and as high as 6 in 10 (60.7%) of the respondents have a history of diabetes, hypertension, or heart disease, personally or in their family, respectively. This agrees with previous studies in the country reporting a prevalence of up to 11% for diabetes [61] and as high as 28.7% for hypertension [62]. As a result of unhealthy eating habits and other lifestyle practices, obesity has also increased in low- and middle-income countries, according to several studies [63]. Nutrition-related NCDs are rising as a result of increased consumption of high energy snacks, processed foods, sedentary lifestyles, and poor-nutrient foods, thus creating an enabling environment [63].

The study has several strengths. This is the first nationwide study performed to assess the dietary changes associated with Ramadan fasting among adult Nigerian Muslims and how these modifications result in body weight changes among the study population. In addition, this study gave us an insight into the prevalence of overweight and obesity in Nigeria. Furthermore, the study also surfaced the prevalence of essential NCDs among Nigerian families, including diabetes, hypertension, and heart disease. Our current research has some limitations. Because the data for this survey were collected online as a self-report, this means that many eligible participants may not be able to participate in it as they may not have access to devices or Internet facilities to do so. Another limitation is reliance on memory for self-reporting of retrospective dietary patterns, weight changes, and perceived health status. Furthermore, since the study was cross-sectional, causality cannot be assigned to Ramadan fasting as an intervening factor. Moreover, social desirability bias may have had a role during this cross-sectional survey. In addition, as most of our respondents were from younger age groups, caution should be exercised when generalizing the findings.

## 6. Conclusions

This study highlighted that Ramadan fasting was associated with extensive dietary modifications that are characterized by a healthy diet and eating habits. These changes are also reflected in changes in body weight as well as a remarkable improvement in the perceived health status of those observing this religious ritual. Public health experts and policymakers should reinforce healthy behaviors during the month of Ramadan and continue through the year so that there is motivation to continue having a healthy diet and lifestyle. Stakeholders should develop strategies that are targeted at bringing nutritional education and interventions to the doorstep of the general public so that the positive effects on weight and lifestyle changes during Ramadan are maintained even when the fasting is completed.

## Figures and Tables

**Figure 1 ijerph-19-14340-f001:**
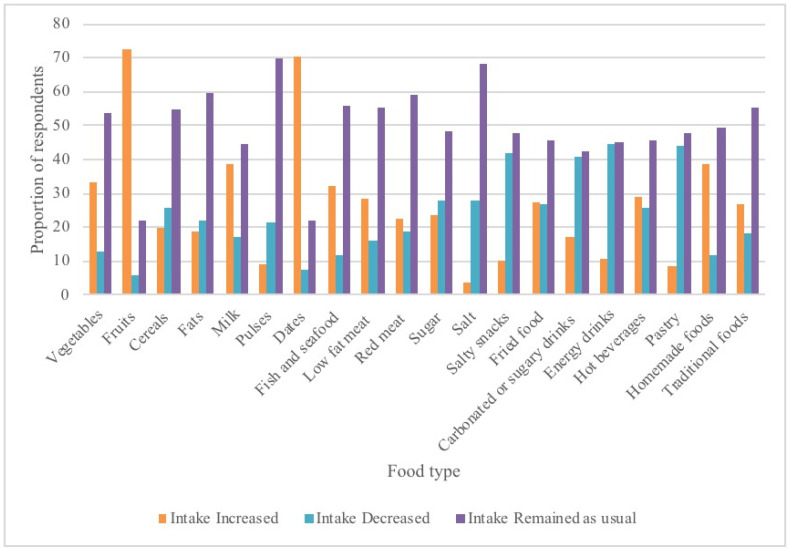
Food groups modified during Ramadan.

**Figure 2 ijerph-19-14340-f002:**
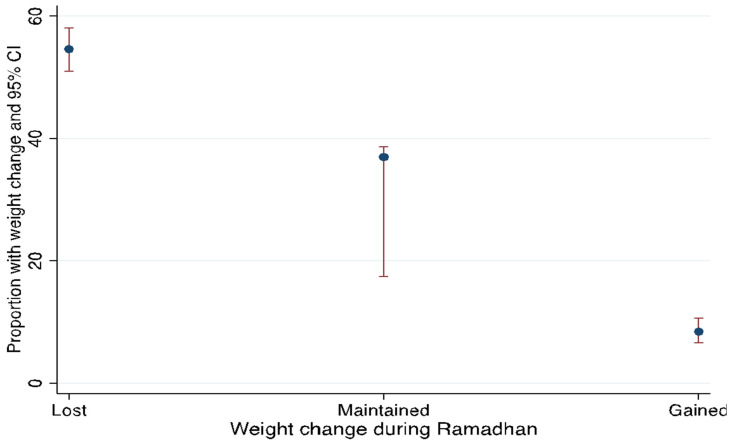
Respondent’s weight change during Ramadan.

**Table 1 ijerph-19-14340-t001:** Background characteristics of study participants.

Variable	Frequency (%), N = 770
Place of permanent domicile	
Urban	562 (73.0)
Rural	208 (27.0)
Sex	
Men	86 (37.1)
Women	484 (62.9)
Age group (Years)	
Less than or equal to 20	76 (9.9)
21–30	476 (61.8)
31–40	191 (24.8)
41–50	24 (3.1)
51–60	3 (0.4)
Marital status	
Single	510 (66.2)
Married	249 (32.3)
Divorced	8 (1.0)
Widowed	3 (0.4)
Highest educational level	
Tertiary or higher	164 (21.3)
Undergraduate	508 (66.0)
Secondary	95 (12.3)
Primary	3 (0.4)
Occupation	
Employed	371 (48.2)
Unemployed	399 (51.8)
* Monthly family income	
Less than NGN 137,600	250 (32.4)
NGN 137,600 or more	231 (30.0)
Do not know	289 (37.5)
BMI (kg/m^2^)	
Normal (<18.5 kg/m^2^)	271 (35.2)
Underweight (18.5–24.9 kg/m^2^)	78 (10.1)
Overweight (25–30) kg/m^2^	113 (14.77)
Obese (>30 kg/m^2^)	186 (24.2)
Do not know	122 (15.8)
History of diabetes, hypertension, or heart disease	
Yes	98 (12.7)
No	672 (87.3)
Family history of diabetes, hypertension, or heart disease	
Yes	467 (60.7))
No	303 (39.3)
Days fasted	
None	21 (2.7)
1–10	5 (0.7)
11–20	105 (13.6)
21–30	639 (83.0)
Frequency of snacking from Iftar to Suhoor	
1 to 2 times daily	294 (38.2)
3 or more times daily	86 (11.2)
1 to 6 times weekly	218 (28.3)
Less than once weekly	172 (22.3)
Modification of diet during Ramadan	
No	123 (16.0)
Yes, a little	245 (31.8)
Yes, moderately	270 (35.1)
Yes, a lot	132 (17.1)

* Monthly living wage for families in Nigeria (2020) = NGN 137,600.00.

**Table 2 ijerph-19-14340-t002:** Diet quality and eating behaviors before and during Ramadan.

Variable	Before Ramadan	During Ramadan	*p*-Value
Self-evaluated quality of diet			
Good	684 (88.8)	716 (92.9)	0.005
Poor	86 (11.2)	54 (7.1)	
Cups of water drunk/day			
1–3	183 (23.8)	223 (29.0)	0.03
4–7	360 (46.8)	314 (40.8)	
8 or more	227 (29.5)	233 (30.3)	
Frequency of ordering takeaway food			
Never or rarely	378 (49.1)	507 (65.8)	<0.001
Daily	24 (3.1)	17 (2.2)	
1–2 times weekly	60 (7.8)	55 (7.2)	
3–6 times weekly	67 (8.7)	37 (4.8)	
1 or more times monthly	241 (31.3)	154 (20.0)	
Frequency of eating out			
Never or rarely	354 (46.0)	581 (75.5)	<0.001
Daily	33 (4.3)	10 (1.3)	
1–2 times weekly	63 (8.2)	24 (3.1)	
3–6 times weekly	73 (9.5)	23 (3.0)	
1 or more times monthly	247 (32.1)	132 (17.1)	
Snacking			
Less often	485 (63.0)	540 (70.0)	<0.001
More often	173 (22.5)	67 (8.7)	
No	112 (14.6)	163 (21.2)	
Consuming large quantities of food			
Less often	200 (26.0)	198 (25.7)	0.27
More often	437 (56.8)	461 (59.9)	
No	133 (17.3)	111 (14.4)	
Eating despite not feeling hungry			
Less often	219 (28.4)	238 (30.9)	0.001
More often	414 (53.8)	445 (57.8)	
No	137 (17.8)	87 (11.3)	

**Table 3 ijerph-19-14340-t003:** Multinomial logistic regression showing crude and adjusted ORs of risk factors associated with weight loss and gain during Ramadan, N = 627.

Variable	Lost Weight	Gained Weight	*p*-Value	Lost Weight	Gained Weight	*p*-Value
	Crude OR (95% CI)	Crude OR (95% CI)		* Adjusted OR (95% CI)	* Adjusted OR (95% CI)	
Place of permanent domicile						
Urban	Reference		0.98			
Rural	1.1 (0.7–1.4)	1.0 (0.5–1.8)				
Age (years)						
20 or less	Reference		0.51	Reference	Reference	
21–30	0.6 (0.4–1.1)	0.7 (0.3–1.8)		0.8 (0.4–1.4)	0.8 (0.3–2.4)	
31–40	0.9 (0.5–1.5)	0.7 (0.3–2.0)		1.0 (0.5–2.1)	0.9 (0.3–3.4)	0.47
41–50	1.3 (0.5–3.9)	1.1 (1.8–6.7)		1.6 (0.5–5.3)	1.6 (0.2–12.1)	
51–60	1.0 9 (0.1–11.6)	--		1.4 (0.1–16.5)	---	
Sex						
Men	Reference		0.35	Reference	Reference	
Women	0.9 (0.7–1.2)	1.3 (0.8–2.3)		0.9 (0.6–1.2)	1.1 (0.5–2.1)	0.43
Occupation						
Employed	Reference		0.67			
Unemployed	1.0 (0.7–1.3)	1.2 (0.7–2.1)				
Pre-Ramadan BMI (kg/m^2^)						
Normal	Reference		0.33	Reference	Reference	
Underweight	0.7 (0.4–1.2)	0.3 (0.1–1.1)		0.7 (0.4–1.2)	0.3 (0.1–1.20	0.19
Overweight	1.1 (0.7–1.8)	1.4 (0.6–3.1)		1.0 (0.6–1.7)	1.3 (0.6–3.1)	
Obese	0.9 (0.6–1.3)	1.0 (0.5–2.1)		0.9 (0.6–1.3)	1.0 (0.5–2.2)	
Days fasted						
None	Reference		0.24			
1–10	0.3 (0.1–2.7)	--				
11–20	1.9 (0.7–5.)	1.0 (0.3–4.5)				
21–30	1.5 (0.6–3.7)	0.6 (1.7–2.4)				
Improved self-evaluated diet quality during Ramadan						
No	Reference		0.97			
Yes	1.0 (0.6–1.8)	0.9 (0.3–2.5)				
Cups of water drunk/day during Ramadan						
1–3	Reference		0.03			0.05
4–7	0.7 (0.3–1.0)	0.5 (0.3–0.9)		0.7 (0.5–1.0)	0.6 (0.3–1.1)	
8 or more	1.0 (0.7–1.5)	0.5 (0.3–1.1)		1.0 (0.7–1.7)	0.5 (0.2–1.0)	
Frequency of ordering food or takeaway during Ramadan						
Never or rarely	Reference		0.01			0.04
Daily	2.9 (0.8–10.2)	1.8 (0.2–18.2)		3.5 (0.8–14.9)	3.0 (0.3–35.3)	
1–2 times weekly	1.0 (0.5–1.8)	3.4 (1.5–7.8)		1.2 (0.6–2.4)	3.7 (1.4–9.8)	
3–6 times weekly	1.7 (0.8–3.7)	3.1(1.0–9.7)		1.7 (0.7–4.1)	1.9 (0.5–7.5)	
1 or more times monthly	0.7 (0.5–2.1)	1.2 (0.6–2.3)		0.7 (0.5–1.1)	0.9 (0.4–2.0)	
Frequency of eating out during Ramadan						
Never or rarely	Reference		0.08			0.34
Daily	1.6 (0.4–6.1)	-		0.8 (0.2–4.1)	--	
1–2 times weekly	0.5 (0.2–1.2)	1.3 (0.4–4.8)		0.5 (0.2–1.4)	0.7 (1.4–3.0)	
3–6 times weekly	1.6 (0.6–4.7)	6.2 (1.8–21.6)		1.3 (0.4–4.3)	4.8 (1.0–20.7)	
1 or more times monthly	1.0 (0.7–1.5)	1.5 (0.8–3.0)		1.1 (0.7–1.7)	1.3 (0.6–2.8)	
Snacking during Ramadan						
No	Reference		0.02			<0.001
Less often	1.3 (.0–1.8)	2.5 (1.1–5.5)				
More often	0.7 (0.4–1.2)	1 (0.3–3.4)				
Consuming large quantities of food during Ramadan						
No	Reference		0.17	Reference		0.04
Less often	1.2 (0.9–1.7)	1.8 (0.9–3.5)		1.4 (0.9–2.0)	2.6 (1.2–5.9)	
More often	0.9 (0.5–1.4)	0.8 (0.3–2.1)		0.7 (0.4–1.2)	0.8 (0.2–3.1)	
Eating despite not feeling hungry during Ramadan						
No	Reference		0.61			
Less often	1.1 (0.8–1.5)	1.7 (0.9–3.2)				
More often	1.0 (0.6–1.7)	1.3 (0.5–3.5)				
Frequency of snacking from Iftar to Suhoor						
Less than once weekly	Reference		0.18			
1 to 2 times daily	0.7 (0.5–1.1)	1.2 (0.5–2.8)				
3 or more times daily	0.8 (0.4–1.3)	2.5 (0.9–6.6)				
1 to 6 times weekly	0.8 (0.5–1.2)	1.7 (0.8–4.1)				

* Adjusted for age, sex, pre-Ramadan BMI, frequency of drinking water, takeaway, eating out, and snacking during Ramadan.

**Table 4 ijerph-19-14340-t004:** Binary logistic regression showing crude and adjusted ORs for factors associated with health state during Ramadan.

Variable	Crude OR (95% CI)	*p*-Value	Adjusted OR * (95% CI)	*p*-Value
Weight change during Ramadan				
Maintained	Reference	0.42		
Lost	0.5 (0.2–1.5)			
Gained	0.6 (0.1–3.0)			
Sex				
Men	Reference	0.15	Reference	0.81
Women	0.5 (0.2–1.3)		0.9 (0.3–2.5)	
Age (Years)				
20 or less	Reference	0.57	Reference	0.64
21–30	2.2 (0.7–6.8)		2.7 (9.7–9.6)	
31–40	2.6 (0.6–10.7)		1.6 (0.2–9.9)	
41–50	1.3 (0.1–12.0)		0.9 (9.1–12.2)	
51–60	--			
Place of permanent domicile				
Urban	Reference	0.7		
Rural	1.2 (0.4–3.3)			
Occupation				
Employed	Reference	0.06		0.09
Unemployed	0.4 (0.2–1.1)		0.5 (0.2–1.1)	
BMI (kg/m^2^)				
Normal	Reference	0.60	Reference	0.74
Underweight	0.7 (0.2–3.8)		0.8 (0.2–4.5)	
Overweight	0.5 (0.2–1.9)		0.5(0.2–2.0)	
Obese	0.5 (0.2–1.5)		0.5 (0.1–1.6)	
History of DM, hypertension, or heart disease				
No	Reference	0.16		
Yes	0.5 (0.2–1.3)			
How many days fasted				
None		0.31		
1–10	0.1 (0–1.1)			
11–20	1 (0.1–9.0)			
21–30	2.4 (0.3–19.3)			
Changed diet quality during Ramadan				
No	Reference	0.05		0.08
Yes	3.3 (1.1–10.1)		2.9 (0.9–9.0)	
Cups of water drunk/day during Ramadan				
1–3	Reference	0.45		
4–7	1.3 (0.5–3.3)			
8 or more	2.1 (0.6–7.2)			
Frequency of ordering food or takeaway during Ramadan				
Never or rarely	Reference	0.81		
Daily	--			
1–2 times weekly	1.8 (0.2–13.5)			
3–6 times weekly	--			
1 or more times monthly	1.2 (0.4–3.7)			
Frequency of eating out during Ramadan				
Never or rarely	Reference	0.18		
Daily	0.2 (0.1–1.5)			
1–2 times weekly	0.2 (0.1–1.0)			
3–6 times weekly	0.4 (0.1–3.4)			
1 or more times monthly	0.4 (0.2–1.1)			
Snacking during Ramadan				
No	Reference	0.10	Reference	0.20
Less often	0.2 (0.1–1.4)		0.2 (0.1–1.4)	
More often	0.2 (0.1–2.3)		0.2 (0.1–2.3)	
Consuming large quantities of food during Ramadan				
No	Reference	0.16		
Less often	0.3 (0.1–1.3)			
More often	0.4 (0.1–2.2)			
Eating despite not feeling hungry during Ramadan				
No	Reference	0.20		
Less often	0.4 (0.1–1.3)			
More often	0.4 (0.1–1.8)			
Frequency of snacking from Iftar to Suhoor				
1 to 2 times daily	Reference	0.23		
3 or more times daily	0.6 (01–3.2)			
1 to 6 times weekly	0.3 (0.1–1.1)			
Less than once weekly	0.4 (0.1–1.4)			

* Adjusted for age, sex, pre-Ramadan BMI, occupation, diet quality, and snacking during Ramadan.

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
