# Peer review of "Nigerian Muslim’s Perceptions of Changes in Diet, Weight, and Health Status during Ramadan: A Nationwide Cross-Sectional Study"

_ijerph, 2022, doi:10.3390/ijerph192114340_

Round 1

Author Response

Response to Reviewer 1 Comments

Journal: International Journal of Environmental Research and Public Health

Manuscript ID: ijerph-1935973

Title: Perceived diet, body weight and changes in health status during the fasting month of Ramadan

among Nigerian adults: A cross-sectional study

Brief Summary

The aim of this study was to assess -reported body

weight changes and health state during Ramadan. This nationwide cross-sectional study was conducted among Nigerian adults using a web-based structured questionnaire. Results showed that Ramadan fasting in Nigeria was associated with improved dietary quality, eating behaviors, and weight loss.

Conclusions highlighted the need for public health measures to support the positive dietary changes

observed during Ramadan so that healthy habits continue throughout the year.

General comments

This study addresses a gap in the knowledge on the impact of Ramadan intermittent fasting on

health and dietary related behaviors among Nigerian adults during the COVID-19 pandemic.

While generally well-written, here and there in the manuscript, a better explanation of some

terminology would be helpful for readers not familiar with the Muslim observance of Ramadan.

The editor's and peer-reviewers comments are greatly appreciated. This manuscript has been improved by the peer-reviewers comments. On a point-by-point basis, we have addressed the pertinent points raised by the reviewers 

This will be noted in specific comments below.

It would be very helpful for non-Muslim readers of this paper to explain the specific

requirements of adherence to Ramadan, such as the time when eating is allowed and if drinking

water is allowed during the day.

Thank you for your suggestions. Lines 28-30 have been edited accordingly for better clarity.

The Methods section lacks many important details. The authors are strongly advised to refer to

the

manuscript.

Specific comments

Introduction

Rationale and objectives for the study are clearly and adequately explained.

Many thanks for the positive comments to emphasize we have followed STROBE Checklist

Methods

Lines 70-71: Why were these dates selected for the beginning and end of the study period (27th of

Ramadan and 29th of Shawwal)? It is unclear if some participants completed the survey tool during

Ramadan and others after the Ramadan fasting period. If participants completed surveys at different

time periods, please explain the rationale for this. Referencing memory after the Ramadan period had

ended should be included as a limitation in the Discussion

My sincere thanks go out to you for the comment.

A single cross-sectional survey was conducted from the end of Ramadan until 4 weeks later. We conducted the survey at the end of Ramadan to assess how diet and lifestyles had changed during the four weeks of the holy month. Taking the reviewers' suggestions into account, we have added limitations regarding self-reporting memory ( Line 279-Line 280).    

Line 83: More detail is needed on how exactly the convenience sample was selected (e.g., referral or

self-selection through advertisements, etc.) Give the eligibility criteria, and the sources and methods of selection of participants. See item 6a on the STROBE Checklist and Explanation and Elaboration.

Lines 83-85: Here the authors refer to the webque earlier (Abstract Lines 10-12 and Methods Lines 70-

72) they refer to a singular web-based questionnaire/survey. Please reconcile this discrepancy. If a

variety of different questionnaires was used, they should be listed and cited individually either in the

text or in a supplementary table.

My sincere thanks go out to you for the comment. Line 76- 80 A convenience and snowball sampling method were employed. Data were collected anonymously with no identifying information using single pre-tested, structured, self-administered, web-based, electronic questionnaires from previous similar studies[32–35]. The survey was created from previous validated questionnaires [32–35] and distributed through social media platforms and email through a unique Google Form link (docs.google.com/forms). We have amended and edited the text accoridngly.

Lines 87-90:

 0.80. Please name and

Lines 92-95: Please specify which outcome was used for the power calculation for the required sample

size for this study. All reported outcomes should be listed with the estimated power to show that the

authors used the most conservative power calculation. Please address items 7-12 of the STROBE

Many thanks for the comments we have mentioned the alpha.1% indicating a 10% chance that a significant difference is actually due to chance and is not a true difference and the B error 0.01 indicating a indicating a 1% chance that a significant difference is missed

Checklist.

Lines 97-99: Please address the comment on Lines 83-85.

Yes the line has been addressed

Lines 142-143: Please state how many were included in the analyses out of the total surveyed.

We included all of the 770 in the analysis

Lines 148-151: Were sex, BMI, and age the only a priory confounding variables?

Yes they were the only priory confounding variables included in the analysis, as seen in the previous literature

Results

Suggested to refer to STROBE Checklist items 13-17

Line 165: Please defi Iftar to Suhoor

Line 29-31 . It has been defined

Lines 186-189: There should be some explanation here that drinking water during the day is prohibited

during Ramadan.

   Water abstinence during fasting was explained in the introduction Line 30 and thus is not repeated here

Figure 2: This figure adds no additional information than what is provided in the text so should be

deleted. If the authors could provide a figure with the amounts of weight lost, etc. that would be more

informative.

Many thanks for the comments and the explanation Figure 2 outlines the weight change and health state during Ramadan. Lines 175 and 178 provide an explanation of Figure 2. Discussion

Lines 250-

Line 251: P-values belong in the Results not the Discussion section. Please remove.

Many thanks, P value has been removed as per suggestions

Lines 255-

, as this was quite significant, especially family income. How does one not

know? It is more likely that participants did not want to report. This information should have been

addressed and described in the statistical analyses as missing data (not at random, etc.).

All respondents gave a response on income i.e none was missing. An option given to them was " Dont know" was available on the questionnaire.

Lines 266-268: This is great information regarding Ramadan being a period of almsgiving, as helps

explain respondents reporting a better quality diet despite the fact that almost a third of them were

below the level of the minimum gross family income.

Many thanks for the comments

Lines 278-279: Please provide some explanation and references as to why some respondents reported

We have edited and amended lines 277 – 279

Lines 322-324: Why do the authors write that a younger age group is less susceptible to social

desirability bias? One could argue that that is exactly when social desirability bias is more likely to occur.

Please delete this statement or provide a reference to back it up.

We have deleted the statement and many thanks for the comments

Conclusion

Well done\

Many Thanks

Reviewer 2 Report

Reviewer comments and suggestions

The authors of this study assessed the Nigerian Muslims' dietary modifications during Ramadan and their associated body weight changes and self-reported health state. 

For this, they conducted a cross-sectional study among Nigerian adult Muslims through a web-based structured questionnaire. 

Sociodemographic, dietary habits, eating behaviors, food choices, perceived weight changes, and health status was assessed. 

The study included 770 participants of which 62.9% were females, with the age range of 18-60 years and a mean age of 27.7± 6.4 years). The result of this showed that the intakes of fruits, palm dates, homemade foods, milk products, and vegetables increased. Compared to the pre-fasting intakes, there was a reduced intake of energy drinks, pastries, salty snacks, and carbonated or sugary drinks during Ramadan. Finally, the study concluded that nearly all (97.3%, 95% CI: .95.8%- 8.3%) reported having good health during Ramadan, and 2.7% (95% CI: 1.7-4.1%) reported having a poorer health state during Ramadan. 

Overall, the manuscript was well written. However, a few concerns/comments needed to be explained.

  1. Line 5-6 Please add the affiliations of the authors
  2. Line 63-64 Before and during the pandemic few concerns have been needed to mention in the introduction as well.
  3. Line 125 What is 47 here? typoerror
  4. How did the authors calculate the intakes, did they know the baseline values
  5. Line 276 Please report other studies that showed similar observation
  6. All references should be modified based on the mdpi journals.

Author Response

Reviewer comments and suggestions

The authors of this study assessed the Nigerian Muslims' dietary modifications during Ramadan and their associated body weight changes and self-reported health state. 

Overall, the manuscript was well written. However, a few concerns/comments needed to be explained.

Thank you to the peer reviewers for their valuable comments. The feedback has improved the quality of the manuscript

  1. Line 5-6 Please add the affiliations of the authors

Many thanks we have added the necessary affiliations

  1. Line 63-64 Before and during the pandemic few concerns have been needed to mention in the introduction as well.

Many thanks we have included in line 59-61

  1. Line 125 What is 47 here? typo error

Many thanks the error has been rectified

  1. How did the authors calculate the intakes, did they know the baseline values?

The study is a self-reported cross-sectional study, as indicated in the methodology. Water consumption was self-reported rather than measured.

  1. Line 276 Please report other studies that showed similar observation

We have amended and edited and written accordingly

  1. All references should be modified based on the mdpi journals.

All references have been modified as per MDPI requirements

Reviewer 3 Report

This research article is interesting since it is the first study that gives data on the diet during Ramadan in Nigerian adults. Some minor comments:

Introduction. As you explain what is Suhoor, please also explain what is Iftar. It appears in Results without mentioned it before.

Methods. Line 125, just cite the reference 47 in the same way.

Table 2. Cite that numbers refer to frequencies (%).

Discussion. Line 227, a space is needed between by and 38.4%

Author Response

We sincerely appreciate the peer -reviewer comments which we have found to enrich the manuscript

This research article is interesting since it is the first study that gives data on the diet during Ramadan in Nigerian adults. Some minor comments:

Many thanks for the valuable comments which we appreciate

Introduction. As you explain what is Suhoor, please also explain what is Iftar. It appears in Results without mentioned it before.

Iftar has been explained in the introduction Line 32

Methods. Line 125, just cite the reference 47 in the same way.

We have amended as per the feedback

Table 2. Cite that numbers refer to frequencies (%).

Many thanks table 2 has been cited in Line 167

Discussion. Line 227, a space is needed between by and 38.4%

Many thanks the space has been added

Round 2

Reviewer 1 Report

Review Report 2

Journal: International Journal of Environmental Research and Public Health

Manuscript ID: ijerph-1935973

Title: Perceived diet, body weight and changes in health status during the fasting month of Ramadan among Nigerian adults: A cross-sectional study

General Comment: While the authors have addressed most of this reviewer’s previous comments satisfactorily, a few specific items still need attention.

Specific Comments:

Line 46: “…with a mean age of 27±76.4 years.” This standard deviation seems unlikely. Please correct this typo.

Line 101: Per this reviewer’s request from the previous review round, “Shawwal” needs to be defined.

Lines 101-102: Please clearly state that these dates correspond to the end of Ramadan and the following 4 weeks, to make it clear to the reader that the survey period was not during Ramadan.

Lines 120-123: This description of the power analysis remains unchanged. The authors still have not addressed the question as to which outcome measure the power calculation was based on (weight change, dietary intake, etc.?). Main outcome measures should be listed with the estimated power for each one to show that the most conservative power estimate was used to estimate the sample size.

Lines 171-172: The authors state in their response that sex, BMI, and age “…were the only confounding variables included in the analysis, as seen in the previous literature”. Please add citations from that previous literature here to back up that statement.

Lines 250-252: The authors have not addressed the reviewer’s comment from the first-round review:

The authors are overstating the results regarding consumption of protein rich foods. Instead of writing “…by most of the respondents”, please say “…by slightly more than half of the respondents.”

Author Response

Many thanks for the peer -reviewers’ comments . We have edited  the minor comments as per the recommendation on a point – point basis

  1. Line 46: “…with a mean age of 27±76.4 years.” This standard deviation seems unlikely. Please correct this typo.
    This has been corrected to 27.±76.4 years.

    2. Line 101: Per this reviewer’s request from the previous review round, “Shawwal” needs to be defined.
    I don't understand what he means here. Defined in terms of what? If it is related to Q.3 that has been addressed

 Many thanks for the comments . for clarity we have removed the  Islamic months here and  added “These dates correspond to the end  Ramadan and the following four weeks after Ramadan”

3. Lines 101-102: Please clearly state that these dates correspond to the end of Ramadan and the following 4 weeks, to make it clear to the reader that the survey period was not during Ramadan.
"Dates correspond to the end of Ramadan and the following 4 weeks" added to the paragraph.
Many thanks for the comments . for clarity we have added  as per the peer -reveiwers recommnedations“These dates correspond to the end  Ramadan and the following four weeks after Ramadan”

  1. Lines 120-123: This description of the power analysis remains unchanged. The authors still have not addressed the question as to which outcome measure the power calculation was based on (weight change, dietary intake, etc.?). Main outcome measures should be listed with the estimated power for each one to show that the most conservative power estimate was used to estimate the sample size.

As there are no specific studies related to changes in weight and perceived health state  during Ramadan in Nigeria we used a standardized priori G* power software and  sample size was calculated using a two-tail student t-test bivariate correlation analysis model to identify the required sample size. For an effect size of 0.15, an alpha error of .1, and a power of 0.99, 687 participants were estimated . As there are limited studies in the context of  lifestyle changes during Ramadan during Nigerian context we used  Our two primary outcomes are weight loss and perceived health state. We have recomputed using  the scanty available literature that is  close to our primary aim.

Fisher's formula was used to estimate  the sample size for this study[35]. Based on the primary outcomes of weight change and health status, two estimates were made. Based on the proportion of respondents who lost weight during Ramadan in a previous study, [36] 95% confidence level, and 5% margin of error, a minimum sample size of 363 was obtained. A 10% increase was made in order to adjust for non-responses, and 400 were rounded up. In the second sample size calculation, we used the percentage of respondents who were satisfied with their health status (30.6%) from the same study,[37] 95% confidence level, and a 5% error margin. We estimated a minimum sample size of 384, which was increased by 10% to account for non-responses and rounded up to 423.

5.Lines 171-172: The authors state in their response that sex, BMI, and age “…were the only confounding variables included in the analysis, as seen in the previous literature”. Please add citations from that previous literature here to back up that statement.

Citation added to the statement and reference added to the list. (Osman F, Haldar S, Henry CJ. Effects of Time-Restricted Feeding during Ramadan on Dietary Intake, Body Composition and Metabolic Outcomes. Nutrients 2020;12:2478. https://doi.org/10.3390/nu12082478.)
6.Lines 250-252: The authors have not addressed the reviewer’s comment from the first-round review:The authors are overstating the results regarding consumption of protein rich foods. Instead of writing “…by most of the respondents”, please say “…by slightly more than half of the respondents.”

Line 259 This has been corrected to "slightly more than half" as recommended by the reviewer.
